



# Quantifying the effects of mixing state on aerosol optical properties

Yu Yao[1], Jeffrey H. Curtis[2], Joseph Ching[3,4,5], Zhonghua Zheng[6,7,8], and Nicole Riemer[1]

[1]Department of Atmospheric Sciences, University of Illinois Urbana-Champaign, Urbana, IL, 61801, USA
[2]Department of Mechanical Science and Engineering, University of Illinois Urbana-Champaign, Urbana, IL, 61801, USA
[3]Meteorological Research Institute, Japan Meteorological Agency, Tsukuba, Ibaraki, 305-0052, Japan
[4]National Institute of Polar Research, 10-3 Midori-cho, Tachikawa, Tokyo, 190-8518, Japan
[5]Research Institute for Humanity and Nature, 457-4 Motoyama, Kamigamo, Kita-ku, Kyoto, 603-8047, Japan
[6]Computational and Information Systems Laboratory, National Center for Atmospheric Research, Boulder, CO, 80307, USA
[7]Climate and Global Dynamics Laboratory, National Center for Atmospheric Research, Boulder, CO, 80307, USA
[8]Advanced Study Program, National Center for Atmospheric Research, Boulder, CO, 80307, USA

**Correspondence:** Nicole Riemer (nriemer@illinois.edu) and Yu Yao (yuyao3@illinois.edu)

**Abstract.** Calculations of the aerosol direct effect on climate rely on simulated aerosol fields. The model representation of aerosol mixing state potentially introduces large uncertainties into these calculations, since the simulated aerosol optical properties are sensitive to mixing state. In this study, we systematically quantified the impact of aerosol mixing state on aerosol optical properties using an ensemble of 1800 aerosol populations from particle-resolved simulations as a basis for Mie calculations for optical properties. Assuming the aerosol to be internally mixed within prescribed size bins caused overestimations of aerosol absorptivity and underestimations of aerosol scattering. Together, these led to errors in the populations' single scattering albedo of up to $-22.3\%$ with a median of $-0.9\%$. The mixing state metric $\chi$ proved useful in relating errors in the volume absorption coefficient, the volume scattering coefficient and the single scattering albedo to the degree of internally mixing of the aerosol, with larger errors being associated with more external mixtures. At the same time, a range of errors existed for any given value of $\chi$. We attributed this range to the extent to which the internal mixture assumption distorted the particles' black carbon content and the refractive index of the particle coatings. Both can vary for populations with the same value of $\chi$. These results are further evidence of the important yet complicated role of mixing state in calculating aerosol optical properties.

## 1 Introduction

Aerosol particles scatter and absorb incoming solar radiation, thereby impacting the global radiative balance and temperatures on Earth (Mitchell Jr, 1971; Charlson et al., 1992; Yu et al., 2006; Winker et al., 2010; Oikawa et al., 2013; Subba et al., 2020). Black carbon (BC), commonly emitted from combustion, has a direct radiative forcing of $+0.9\,\mathrm{W\,m^{-2}}$, which is next only to $CO_2$ (Bond et al., 2013; Gustafsson and Ramanathan, 2016) in its warming impact. At the same time, the overall global average aerosol direct radiative forcing in the clear-sky environment is $-1.9\,\mathrm{W\,m^{-2}}$ because of the presence of other non-absorbing aerosol species, which exert a cooling impact (Bellouin et al., 2005).

Radiative effects of aerosols depend on their optical properties, which, as a whole, are determined by the individual particles that the aerosol consists of. As observed in field campaigns, particles are mixtures of inorganic and organic species, and exhibit




significant spacial and temporal variation in their abundance and composition (Zhang et al., 2007; Bzdek et al., 2012; Laskin et al., 2006), with considerable diversity in composition existing within individual aerosol populations. The topic of this paper is to quantify the importance of diversity in composition for aerosol optical properties.

Aerosol composition impacts aerosol optical properties for several reasons. First, aerosol species differ in their complex refractive index. While inorganic species and many organic species have a purely real refractive index for wavelength of visible sunlight (i.e., only scatter radiation), black carbon and some organic carbon species have a non-zero imaginary part of the refractive index and hence also absorb radiation (Corbin et al., 2018; Esteve et al., 2014; Cappa et al., 2019). Second, aerosol species differ in their hygroscopicity. This governs aerosol water uptake in a humidified environment, which is important for

scattering (Michel Flores et al., 2012; Zieger et al., 2013; Titos et al., 2014, 2016).

Lastly, the arrangement of the different aerosol species within a particle is important for determining their scattering and absorption. For mixed particles without strongly absorbing species, i.e., BC, a volume-mixing rule can be used to calculate the refractive index of the entire particle. When the particle contains BC, assuming a core-shell configuration was proven to be more accurate (Bond et al., 2006) (still assuming sphericity as particle shape). The absorption enhancement of BC-containing

particles due to their non-absorbing coatings has been widely investigated (Moffet and Prather, 2009; Liu et al., 2017; Wu et al., 2020; Fierce et al., 2020). Taking the non-spherical shapes of BC-containing particles into account complicates matter considerably since Mie calculations cannot be applied and more sophisticated optical models need to be used, which are computationally much more expensive. By using the Discrete Dipole Approximation model, Scarnato et al. (2013) found that the absorption coefficients enhancement of BC-NaCl mixtures is higher for compact BC particles completely embedded in

NaCl than for lacy BC particles.

To understand the importance of aerosol composition in calculating aerosol optical properties, it is useful to define the term aerosol mixing state, that is, the distribution of aerosol species among the particles in a population (Riemer et al., 2009; Riemer and West, 2013). Aerosol mixing state in the ambient atmosphere ranges between the two idealized extremes of an external mixture on the one hand, where each particle is composed of a single species, and an internal mixture on the other

hand, where all particles consist of the same mixture of species. Aerosols close to emission sources tend to be more (although not completely) externally mixed (Bondy et al., 2018; Rissler et al., 2014). After emission, aging processes, such as coagulation between particles and condensation of gas species on the particles, transform aerosol populations towards more internal mixtures (Healy et al., 2014; Liu et al., 2013; Zaveri et al., 2010). Past studies quantified the importance of mixing state for aerosol optical properties through optical closure studies. For example, using measured aerosol size distributions and aerosol

composition observed over the East China Sea, Koike et al. (2014) found that the internal mixture assumption for fine particles increased the absorption aerosol optical depth by a factor of 2 or more.

Aerosol mixing state is challenging to represent in 3D chemical transport models, which usually rely on simplifying assumptions for computational efficiency. These assumptions then influence the magnitude of calculated aerosol optical properties. Optical properties are here understood by three widely-used parameters: the absorption cross section, the scattering cross sec-

tion and the asymmetry parameter (Majdi et al., 2020). Many 3D models use a modal approach to represent aerosols, such as the Community Multiscale Air Quality Modeling System (CMAQ)(Binkowski et al., 2007; Appel et al., 2017) and Modal





Aerosol Module (MAM)(Liu et al., 2012). The modes are externally mixed from each other, whereas within each mode, the aerosol is assumed to be internally mixed. For BC-containing modes, sphericity and a core-shell configuration are assumed, so that Mie calculations can be applied to calculate optical properties. Fierce et al. (2016) found that neglecting the diversity in coating thickness for BC-containing particles (a result of the internal mixture assumption) leads to overestimated absorption enhancement by up to 200%. Another approach is the sectional model representation, which tracks size-resolved composition, but not particle composition diversity within a certain size, such as TwO-Moment Aerosol Sectional (TOMAS) and the GLObal Model of Aerosol Processes (GLOMAP)(Kodros et al., 2018; Spracklen et al., 2005). Still, mixing state assumptions need to be invoked for each size bin.

The uncertainties in optical properties introduced by mixing state assumptions were also evaluated through model sensitivity studies. Using the AQMEII-2 model inter-comparison framework, Curci et al. (2015) quantified the sensitivity of aerosol optical properties to several parameters, including aerosol mixing state and size distribution. They found that aerosol mixing state is the dominant factor introducing uncertainties, explaining 30–35% of the uncertainty in aerosol optical depth and single scattering albedo (SSA). Kodros et al. (2018) found that the direct radiative forcing (DRF) can vary from $-1.65$ to $-1.34$ W m$^{-2}$ over the pan-Arctic region depending on the assumption of internal or external mixture. The variation is similar when the assumptions are used to calculate DRF at the top of atmosphere (Ma et al., 2012). These sensitivity studies have in common that no benchmark simulations exist that represent the real mixing state, and therefore the importance of mixing state can only be assessed based on differences between varied idealized assumptions. By applying a detailed particle-resolved benchmark model, Fierce et al. (2017) found that simple mixing state assumptions can result in an erroneous distribution of BC cores and coating material and lead to errors in absorption. This effect was further confirmed to be the main source for the discrepancies between simulated and experimentally-determined particle optical properties (Fierce et al., 2020).

The goal of this study is to systematically quantify the errors in optical properties due to simplified assumptions for mixing state, here quantified with the mixing state metric $\chi$ (Riemer and West, 2013). A similar framework was used to quantify the error in CCN concentration (Ching et al., 2017), showing that CCN error ranges from $-40\%$ to 150% when assuming the aerosol was internally mixed. The error depended on supersaturation level that CCN concentrations were evaluated at, and also aerosol mixing state. In this work, we want to answer the questions: Given the aerosol mixing state, what is the error in aerosol optical values when assuming internal mixture and what are the leading causes for this error?

The paper is structured as follows: Model description, scenario design and the definition of metrics are given in Sect. 2. Section 3 shows the relation between the errors in aerosol scattering and absorption and mixing state for dry aerosol populations, and Sect. 4 further analyzes the errors for the aerosol populations at different levels of ambient relative humidity. The errors in single scattering albedo and its implications for aerosol direct radiative forcing are analyzed in Sect. 5. Section 6 summarizes the main findings.



## 2  Model description, scenario libraries and metrics

### 2.1  The stochastic particle-resolved model PartMC-MOSAIC

The model used for this study is the particle-resolved model PartMC-MOSAIC (Particle Monte Carlo Model-Model for Simulating Aerosol Interactions and Chemistry). A comprehensive description of the model can be found in Riemer et al. (2009) and DeVille et al. (2011, 2019) for PartMC, and in Zaveri et al. (2008) for MOSAIC. PartMC is a Lagrangian box model that tracks the evolution of particles in a fully-mixed computational volume. The processes of emission, coagulation and dilution are simulated stochastically. Gas-phase chemistry and gas-aerosol partitioning are incorporated by coupling with the deterministic

model MOSAIC. Specifically, MOSAIC uses the carbon bond based mechanism CBM-Z for gas-phase photochemical reactions (Zaveri and Peters, 1999), the Multicomponent Taylor Expansion Method (MTEM) for calculating electrolyte activity coefficients in aqueous inorganic mixtures and the Multicomponent Equilibrium Solver for Aerosols (MESA) for calculating the phase states of the particles (Zaveri et al., 2005). The secondary organic aerosol (SOA) treatment follows the Secondary Organic Aerosol Model (SORGAM) (Schell et al., 2001). Aerosol water uptake is calculated using the Zdanovskii-Stokes-

Robinson (ZSR) method (Zaveri et al., 2008; Zdanovskii, 1948; Stokes and Robinson, 1966) based on the composition of the inorganic portion of the particles. By this method, organic species are treated as hydrophobic, and do not contribute to water uptake. The impact of this assumption on optical properties was quantified by Nandy et al. (2021), where they found that errors in single scattering albedo can be up to 6% if neglecting the water uptake of organic compounds.

### 2.2  Scenario library design

Following the strategy in Zheng et al. (2021) and Hughes et al. (2018), we created a scenario library of PartMC-MOSAIC simulations, for this study with a focus on the aging of carbonaceous aerosol. To produce particle populations with a wide range of compositions and mixing states, we varied the model input parameters within the ranges shown in Table 1. We used Latin hypercube sampling (McKay et al., 2000) to create input parameter combinations for a total of 100 model simulations. The simulation time for each simulation was 24 hours beginning at 6 am local time with hourly output. This yielded a total

of 2500 particle populations. All scenarios were run with 10 000 computational particles. To create aerosol initial conditions with realistic mixing states, we adopted the approach described in Zheng et al. (2021): We carried out a first set of simulations, starting with the aerosol initial concentrations set to zero for all simulations (the "initial runs"). We then repeated the same set of simulations, but replaced the aerosol initial condition with a randomly sampled population from the initial runs (the "restart runs"). For the analysis in this paper, we only used the results from the restart runs. Within our ensemble or aerosol

populations, some were found with higher species concentrations than what would be expected in the ambient atmosphere. We applied upper thresholds to eliminate those which were calculated as the sum of the 75[th] percentile and 1.5 IQR (interquartile range) for each of the aerosol species. After this procedure, 1809 out of 2500 populations were used for the error analysis presented in the remainder of the paper.





**Table 1.** Baseline and range for the input variables

| Input parameters | Baseline | Range |
|---|---|---|
| Enviroment Variables | | |
| Relative humidity (RH) | | [0.1, 1) or [0.4, 1) |
| Latitude | | $(70^oS, 70^oN)$ or $(90^oS, 90^oN)$ |
| Day of year | | [1, 365] |
| Temperature | | Based on latitude and day of year |
| Gas emission rates (mol m$^{-2}$ s$^{-1}$) | | |
| Sulfur dioxide (SO$_2$) | $8.5\times 10^{-9}$ | [0-200%] |
| Nitrogen dioxide (NO$_2$) | $3.0\times 10^{-9}$ | [0-200%] |
| Nitrogen oxide (NO) | $5.7\times 10^{-8}$ | [0-200%] |
| Ammonia (NH$_3$) | $8.9\times 10^{-9}$ | [0-200%] |
| Carbon oxide (CO) | $7.8\times 10^{-7}$ | [0-200%] |
| Methanol (CH3OH) | $2.3\times 10^{-10}$ | [0-200%] |
| Acetaldehyde (ALD2) | $1.7\times 10^{-9}$ | [0-200%] |
| Ethanol (ANOL) | $5.3\times 10^{-9}$ | [0-200%] |
| Acetone (AONE) | $7.8\times 10^{-10}$ | [0-200%] |
| Dimethyl sulfide (DMS) | $3.8\times 10^{-11}$ | [0-200%] |
| Ethene (ETH) | $1.8\times 10^{-8}$ | [0-200%] |
| Formaldehyde (HCHO) | $4.1\times 10^{-9}$ | [0-200%] |
| Isoprene (ISOP) | $2.4\times 10^{-10}$ | [0-200%] |
| Internal olefin carbons (OLEI) | $5.9\times 10^{-9}$ | [0-200%] |
| Terminal olefin carbons (OLET) | $5.9\times 10^{-9}$ | [0-200%] |
| Paraffin carbon (PAR) | $1.7\times 10^{-7}$ | [0-200%] |
| Toluene (TOL) | $6.1\times 10^{-9}$ | [0-200%] |
| Xylene (XYL) | $5.6\times 10^{-9}$ | [0-200%] |
| Carbonaceous aerosol emission (single mode) | | |
| Geometric mean diameter ($D_g$) | | [25, 250] nm |
| Geometric standard deviation of diameter ($\sigma_g$) | | [1.4, 2.5] |
| BC/OC mass ratio | | [0, 100%] |
| Particle emission flux | | $[0, 1.6\times 10^7]$ m$^{-2}$ s$^{-1}$ |



**Table 2.** Refractive indices of aerosol species at $\lambda = 550$ nm

| Compounds | Refractive index |
|:---:|:---:|
| $H_2SO_4$ | 1.43 |
| $(NH_4)_2SO_4$ | 1.52 |
| $(NH_4)HSO_4$ | 1.47 |
| $NH_4NO_3$ | 1.5 |
| $H_2O$ | 1.33 |
| BC | $1.82 + 0.74i$ |
| SOA | 1.45 |
| OC | 1.45 |

## 2.3 Optical properties calculations

We calculated the optical properties of the particle populations using Mie calculations (Zaveri et al., 2010). These properties included the asymmetry parameter $g$, scattering cross section $\sigma_{\text{scat}}$ and absorption cross section $\sigma_{\text{abs}}$ for each particle. Particles were assumed to be spherical, and when BC was present, a core-shell configuration was assumed, with BC as the core and non-BC species as the shell. In PartMC-MOSAIC, each chemical species was assigned a refractive index and the values were the same as Zaveri et al. (2010), as listed in Table 2. The shell refractive index of the particle was the volume average of all

the shell species, including aerosol water. The absorptivity of brown carbon has been of great interest in recent years (Corbin et al., 2018; Cappa et al., 2019), however, this was not considered in the current work. We used the values for wavelength $\lambda$ of 550 nm for our analysis. In PartMC-MOSAIC, all particles are tracked individually in a well-mixed computational volume, and we obtained the ensemble optical property values by summing over all particles in the volume. The ensemble scattering coefficients $\beta_{\text{scat}}(\lambda)$, ensemble extinction coefficients $\beta_{\text{ext}}(\lambda)$ and ensemble absorption coefficients $\beta_{\text{abs}}(\lambda)$ at wavelength $\lambda$

are given as

$$\beta_{\text{scat}}(\lambda) = \sum_{i}^{N} \sigma_{\text{scat},i}(\lambda) n_i, \tag{1}$$

$$\beta_{\text{ext}}(\lambda) = \sum_{i}^{N} \sigma_{\text{ext},i}(\lambda) n_i, \tag{2}$$

$$\beta_{\text{abs}}(\lambda) = \beta_{\text{ext}}(\lambda) - \beta_{\text{scat}}(\lambda), \tag{3}$$

where $i$ is the particle index, $n_i$ is the number concentration associated with particle $i$ and $N$ is the number of computational

particles in the population. We determined the optical properties of all particle populations of our scenario libraries using these equations.





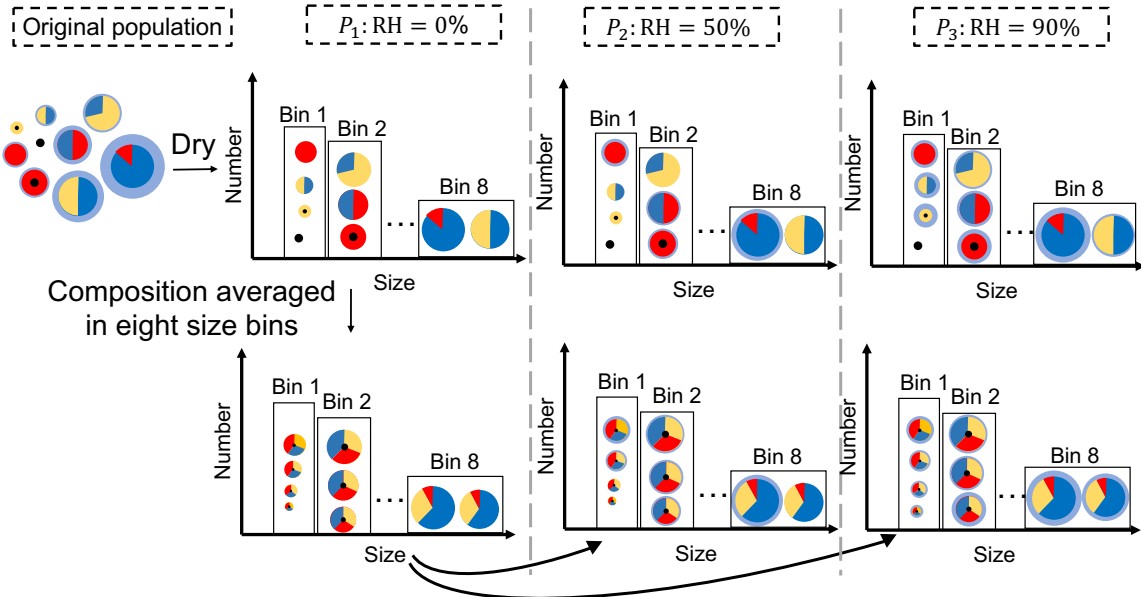

**Figure 1.** Conceptual framework of composition-averaging. The colors indicate different aerosol species. Light blue stands for water, black stands for black carbon black, and other colors are for the other chemical species. Composition averaging is applied to the dry populations, and then water uptake is recalculated for RH = 50% and RH = 90%.

## 2.4 Quantifying the impact of mixing state through composition-averaging

To quantify the impacts of mixing state on aerosol optical properties, we employed the strategy of "composition-averaging" similar to Ching et al. (2016) to create sensitivity scenarios. The technique is shown conceptually in Fig. 1. For each population in our reference scenario library, we averaged the dry particle compositions within prescribed size bins. We chose eight size bins between 0.039 and 10 μm, consistent with the bin structure of the sectional aerosol module MOSAIC used in WRF-Chem (Fast et al., 2006).

The composition-averaging procedure preserves the bulk mass concentration of each species, the total number concentration, and the particle diameters within each bin (Ching et al., 2012). It changes the per-particle compositions so that each bin becomes internally mixed, however the composition can vary between bins. This mimics the assumption frequently made in sectional models, namely that each size bin contains an internally mixed aerosol. PartMC-MOSAIC represents particles outside the MOSAIC bin range, especially for the lower boundary, and we used an extra bin (bin 0) to preserve the total number and mass concentrations. Since the aerosol water content plays an important role for aerosol optical properties, we further calculated water uptake for the reference populations and for the composition averaged populations for 50% ($P_2$) and for 90% ($P_3$) relative humidity, respectively. At $RH = 50\%$, depending on the exact composition, some particles take up water, and at $RH = 90\%$, most particles take up water, except particles that only contain hydrophobic species, such as pure black carbon



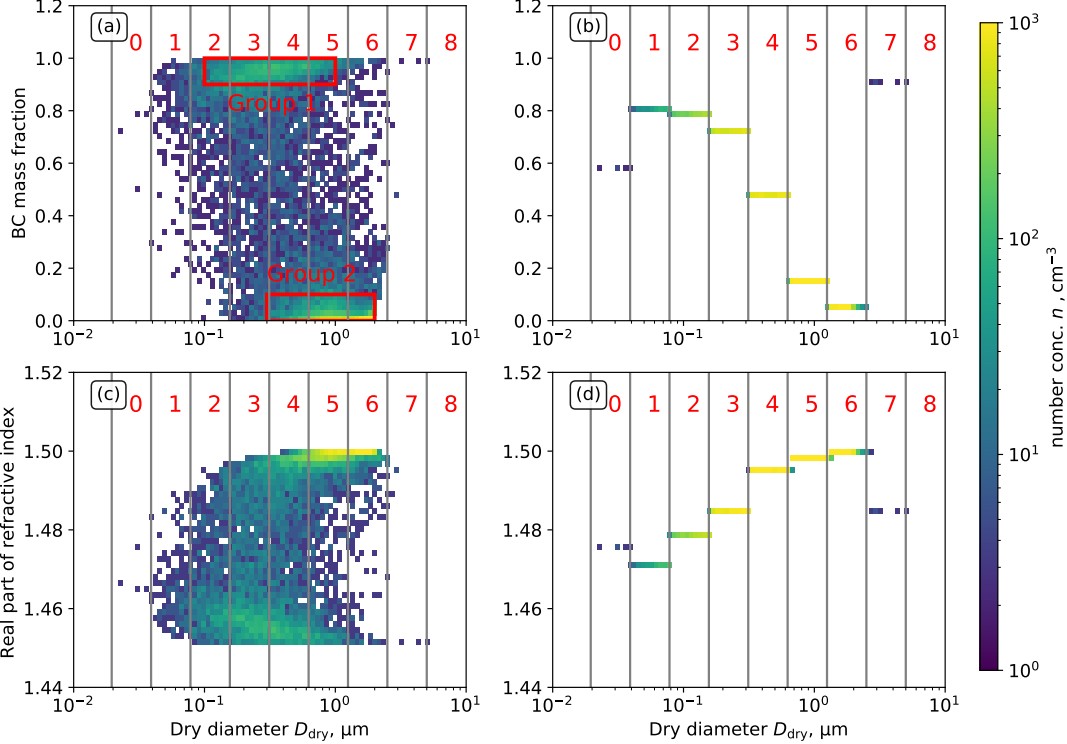

**Figure 2.** Two-dimensional number distributions of BC mass fraction and dry diameter (a (reference), b (composition-averaged)), and real part of the refractive index and dry diameter (c (reference), d (composition-averaged)). The population is taken from scenario 76 at $t = 1$ h, with $\chi = 36\%$. Red numbers and grey vertical lines represent the size bin ranges. The two red rectangles are for the analysis in Sect. 3.

or primary organic carbon. Note that while the dry aerosol mass was conserved by the composition-averaging procedure, the water content was re-calculated after composition-averaging and could change compared to the reference population.

Figure 2 illustrates the changes of two important parameters for aerosol optical properties due to composition-averaging, BC mass fraction and the real part of the refractive index. In the reference case, a wide range of BC mass fractions exists within the same size bin (Fig. 2(a)). After composition-averaging, all particles within a size bin have the same BC mass fraction (Fig. 2(b)). Since composition-averaging preserves the particle diameters, BC and other species are redistributed so that all particles within a size bin are assigned the same mass fractions. Specifically, if a particle has lower BC mass fraction than the average level in the same size bin, BC is added to this particle from those with higher BC content. The coating species are also

redistributed after composition-averaging which causes the refractive index of the coating to change (Fig. 2(c) and (d)). Hence, comparing optical properties before and after composition-averaging in the dry population $P_1$ isolates the impact of mixing state on aerosol optical properties. We will discuss the impact of composition-averaging for dry conditions in Sect. 3 and the impact of water uptake in Sect. 4.





## 2.5 Mixing state metrics

We quantified the optical properties error introduced by a simplified mixing state representation by using the metrics developed by Riemer and West (2013). These metrics include the single-particle diversity $D_i$, the average particle species diversity $D_\alpha$ and bulk population species diversity $D_\gamma$. For a population with $N$ particles, total mass $\mu$ and $A$ species, we can calculate these metrics from the total mass of particle $i$, $\mu_i$, total mass of species $a$ in the population, $\mu^a$, and mass of species $a$ in particle $i$, $\mu_i^a$, for $i = 1, \ldots, N$ and $a = 1, \ldots, A$. The mass fraction of species $a$ in particle $i$, $p_i^a$, mass fraction of particle $i$ in the population, $p_i$ and mass fraction of species $a$ in the population, $p^a$ are given by

$$p_i^a = \frac{\mu_i^a}{\mu_i}, \quad p_i = \frac{\mu_i}{\mu}, \quad p^a = \frac{\mu^a}{\mu}. \tag{4}$$

The single particle diversity $D_i$ describes the effective species number in each particle, and is defined as

$$D_i = \prod_{a=1}^{A} (p_i^a)^{-p_i^a}. \tag{5}$$

For particles containing the same number of species type, particle diversity $D_i$ reaches its maximum when species are present in equal amounts. Based on $D_i$, we can construct $D_\alpha$ and $D_\gamma$, which describes the average effective species number in each particle and bulk population respectively:

$$D_\alpha = \prod_{i=1}^{N} (D_i)^{p_i}, \tag{6}$$

$$D_\gamma = \prod_{i=1}^{A} (p^a)^{-p^a}. \tag{7}$$

Finally, the mixing state metric $\chi$ is defined as the affine ratio of $D_\alpha$ and $D_\gamma$:

$$\chi = \frac{D_\alpha - 1}{D_\gamma - 1}. \tag{8}$$

The values of $\chi$ vary between 0% to 100%. When $\chi = 0\%$, it indicates that the population is fully externally mixed and each particle only contains one species. The population is internally mixed when $\chi = 100\%$, and all particles have the same species mass fractions. For this work, our focus is the optical properties of the particles. Differing from the traditionally used chemical species mixing state index (Riemer and West, 2013; Healy et al., 2014; Bondy et al., 2018; Ye et al., 2018), we grouped the aerosol species by absorbing and non-absorbing species and defined a new index, $\chi_{\text{opt}}$. It still ranges between 0% to 100% and signifies the degree to which absorbing and non-absorbing species are mixed. Since we only consider two (surrogate) aerosol species, the maximum value of $D_i$, $D_\alpha$ and $D_\gamma$ is 2. For the remainder of the paper, we will refer to this metric simply as $\chi$. The same metric was chosen by Yu et al. (2020) to characterize the mixing state of BC-containing aerosol in Beijing and by Zhao et al. (2021) to understand the role of mixing state for aerosol light absorption enhancement.

Figure 3 shows the range of bulk chemical species concentrations, mixing state metric, and optical properties within the selected scenario library. The simulated aerosol bulk species mass concentration in the library covered a wide range of urban





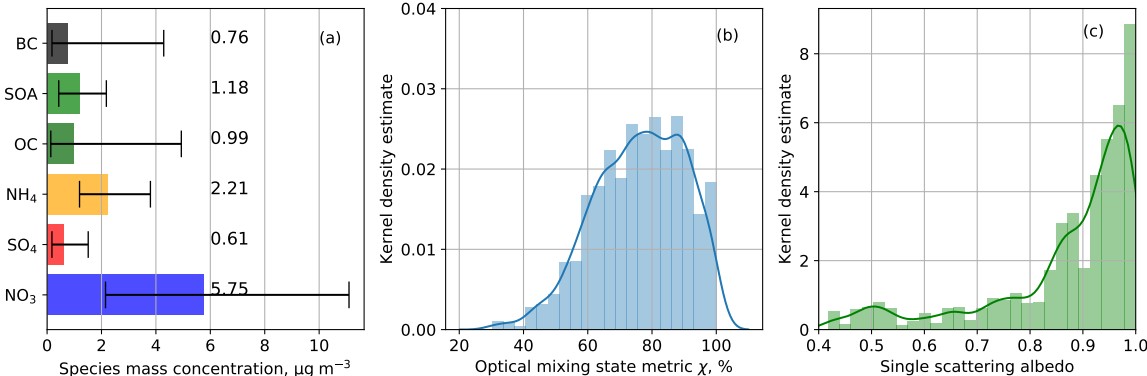

**Figure 3.** Distribution of (a) bulk species concentration, (b) optical mixing state $\chi$ and (c) SSA in the scenario libraries. Error bars in (a) are for $\pm 1$ interquartile range (IQR) and numbers are the species median concentration in $\mu g\,m^{-3}$.

conditions (Fig.3(a)), and the values were comparable to the measurements in different locations (Jimenez et al., 2009; Lanz et al., 2010). Most populations had a mixing state metric $\chi$ larger than 40%, with a median value of 85%. The fact that $\chi$ values smaller than 30% did not occur in our scenario library is consistent with the notion that BC rarely exists in a completely external mixture. Rather, it is frequently co-emitted with organic carbon, which form internal mixtures at the time of emission. Additionally, in urban environments, BC ages quickly, forming internal mixtures with secondary species. Our range of $\chi$ values encompasses the range observed in field measurements at Taizhou, China, where $\chi$ ranged between 68% and 79% for a period in May/June (Zhao et al., 2021), and at Beijing, China, where $\chi$ ranged between 55% and 70% in winter and between 60% and 75% in summer (Yu et al., 2020). Figure 3(c) shows that the single scattering albedo (SSA) was larger than 0.4 for all populations, with a median value of 0.88. While SSA values lower than 0.5 are considered extremely low (4%), most populations (72%) had a SSA larger than 0.85, which is consistent with fine mode SSA observations from AERONET (Levy et al., 2007).

## 3 Errors in aerosol absorptivity and scattering for dry particles

This section describes how we quantified the error introduced by composition-averaging assumptions and how this error depends on mixing state. Similar to the approach used by Ching et al. (2017), we stratified the populations by the optical mixing state metric $\chi$. To isolate the impacts of mixing state (in the sense of how the chemical species except for aerosol water are distributed across the population) from the impacts of water uptake, we first analyzed the results for the dry population scenarios $P_1$. Particles were partially or fully deliquescent in scenarios $P_2$ (RH50) and $P_3$ (RH90). These populations will be further analyzed in Sect. 4 to quantify the water uptake effects on aerosol optical properties resulting from internally mixing hygroscopic and hydrophobic species.





The errors in aerosol optical properties due to the internal mixture assumption were defined by comparing the values of reference and composition-averaged populations. The relative error $\epsilon$ for the aerosol populations was calculated as

$$\epsilon(v, \chi) = \frac{v'(\chi) - v(\chi)}{v(\chi)}, \tag{9}$$

where $v$ stands for $\beta_{\mathrm{abs}}$, $\beta_{\mathrm{scat}}$ or single scattering albedo, and $\chi$ is the mixing state metric.

### 215  3.1  Errors in aerosol absorptivity due to composition-averaging

Absorption was overestimated universally after composition-averaging, and, as expected, the error was higher for more externally-mixed populations (low $\chi$ values), with $\epsilon(\beta_{\mathrm{abs}})$ reaching up to +70% for $\chi$ of 30% (Fig. 4). Each dot in Fig. 4 represents a particle population from the scenario library. As shown in the box plot inset, the mean overestimation was 18% and the maximum reached over 80%. The figure further contains information of BC bulk mass concentration and relative average BC
core size changes, which are the two main factors in determining absorptivity (Bond and Bergstrom, 2006), as represented by marker size and color, respectively. The relative average BC core size change for a population is defined as

$$\Delta D^{\mathrm{core}} = \frac{\sum_{i=1}^{N} n_i D_i^{\mathrm{core}\prime} - \sum_{i=1}^{N} n_i D_i^{\mathrm{core}}}{\sum_{i=1}^{N} n_i D_i^{\mathrm{core}}}, \tag{10}$$

where $i$ is the particle index, $n_i$ and $D_i^{\mathrm{core}}$ are the associated number concentration and core diameter in the reference scenario, and $D_i^{\mathrm{core}\prime}$ is the core diameter in the sensitivity scenario. It is interesting to note that $\Delta D^{\mathrm{core}}$ is always positive, that is, the
average core diameter after composition-averaging is larger than the average core diameter before composition-averaging. This is a result of particle mass being a convex function of particle diameter (assuming spherical particles). Calculating the new core diameters after composition averaging will therefore always lead to on-average larger core diameters than averaging the core diameters before composition averaging.

The decreasing error with increasing $\chi$ can be explained by the magnitude of $\Delta D^{\mathrm{core}}$. Evidently, composition-averaging
caused larger changes of BC core sizes when the populations were more externally mixed. For example, for $\chi = 30\%$, the change in core sizes was as large than +25%, while for $\chi = 95\%$, the change in core sizes was less than 5%. We also noticed a range of errors for populations with $\chi$ between 60 and 70%, i.e., partially internally-mixed populations. In fact, the highest overestimation of 82% was reached at $\chi = 63\%$. As indicated by the circle size, these populations contained very little BC (0.01 µg m$^{-3}$), and even small changes in core sizes can lead to large relative errors in the volume absorption coefficient.
Given the constraint that composition-averaging preserves the particle number concentration and sizes, it follows that, for some particles, this operation increases the sizes of BC cores (while at the same time decreasing the coating thickness), whereas for other particles it decreases the BC cores sizes (while increasing the coating thickness). It is therefore not immediately clear that composition-averaging consistently causes overestimation of aerosol absorption coefficients.

At a per-particle scale, for particles of the same diameter, $\sigma_{\mathrm{abs}}$ increases with increasing BC core, even though the coating
thickness (and hence the absorption enhancement) decreases (Fig. S1). However, $\epsilon(\beta_{\mathrm{abs}})$ is determined by the entire population. The internal mixture in each size bin is reached by moving species from a group of particles to another group of particles. As the BC mass fraction distribution in Fig. 2 shows, there are two major groups of particles in the population: Group 1 are

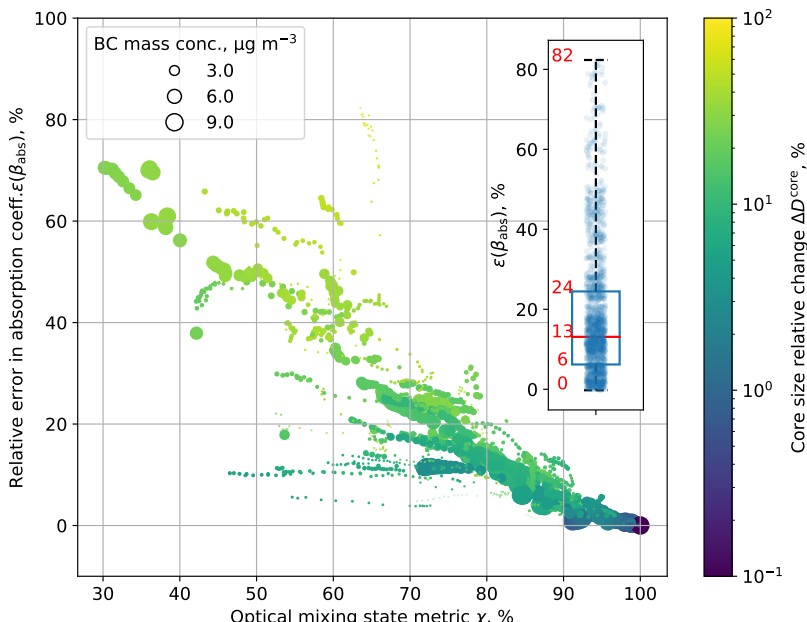

**Figure 4.** Relative error in absorption coefficients $\epsilon(\beta_{\mathrm{abs}})$ after composition averaging for dry particles. Each marker represents an aerosol population. The color denotes the change of BC diameter due to composition-averaging, and the marker size represents BC bulk mass in the population. The box plot inset shows the distribution of the error. The red line shows the median, and the edges of the dashed lines are the minimum and maximum values. Red numbers are for the minimum, first quartile, median, third quartile and maximum values.

particles with higher BC mass fraction, and group 2 are particles with lower BC mass. Particles in group 1 experience decreased absorbing ability because they are losing BC, and vice versa for particles in group 2.

To further illustrate the effects at the population level, we show the effects of composition-averaging on the volume absorption coefficient for a simplified case of five monodisperse populations of different sizes, starting out with completely externally mixed populations consisting of BC and ammonium bisulfate (Fig. 5). Absorption coefficients are normalized by the absorption coefficient for $f_{\mathrm{BC}} = 1$ (pure BC). The black line shows the normalized volume absorption coefficient for populations when all particles are externally mixed for bulk BC mass fractions $f_{\mathrm{BC}}$ varying between 0 and 100%. For external mixtures, absorption

increases linearly with increasing BC mass fraction (black line). The linear relationship applies for all five externally-mixed populations with different diameters, so we can only see one black line in the figure.

The colored lines represent the internally-mixed monodisperse populations (i.e., after composition-averaging) for different diameters. These populations all have higher absorption coefficients compared to the corresponding externally mixed populations. The effect is more pronounced for larger particles and intermediate BC mass fractions because the maximum $\Delta_{\mathrm{core}}$ is

reached. As the table (Fig.5) shows, for a 300 nm population, the normalized absorption is 0.76 when the particles are inter-





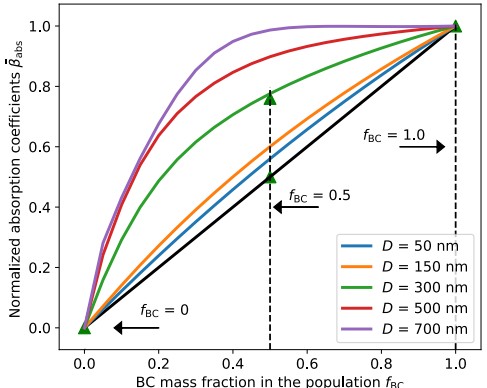

| BC mass Fraction $f$ | External Mixture | Internal Mixture | Normalized $\beta_{\mathrm{abs}}(\mathrm{ext})$ | Normalized $\beta_{\mathrm{abs}}(\mathrm{int})$ |
|---|---|---|---|---|
| 0% | | | 0.0 | 0.0 |
| 50% | | | 0.5 | 0.76 |
| 100% | | | 1.0 | 1.0 |

**Figure 5.** Normalized absorption coefficient as a function of BC mass fraction for five monodisperse populations with different sizes. The coating species is ammonium bisulfate with refractive index 1.47. Absorption coefficients are normalized by $\beta_{\mathrm{abs}}$ of the population with $f_{\mathrm{BC}} = 1$ (pure BC). The black line is for BC in external mixture. Colored lines are for BC in internal mixture of different sizes. Table on the right sketches three 300 nm internal and external populations with BC mass fraction of 0%, 50%, and 100%. Black is for BC and yellow for coating species.

nally mixed, higher than an external mixed population (0.5). Although this example is an idealized case since our populations lie between external and internal mixtures before composition-averaging and are polydisperse, this illustrates that assuming internal mixture will lead to absorption overestimation.

### 3.2 Error in aerosol scattering due to composition-averaging

Considering the volume scattering coefficient, composition-averaging resulted in a negative relative error (Fig.6(a)). Similar to what we found for $\epsilon(\beta_{\mathrm{abs}})$, the magnitudes of $\epsilon(\beta_{\mathrm{scat}})$ decreased with increasing $\chi$, but were overall smaller, with the largest underestimation of $-32\%$ for a population with $\chi = 40\%$ and a median of $-1.2\%$.

Two factors affect the particle scattering ability by composition-averaging, the change of the BC core size (and the corresponding change in coating thickness), and the change in the refractive index of the coating. As Fig. 7 shows, adding a BC

core decreases the scattering ability for particles with diameters less than 1200 nm, which is the typical size range considered in our study. This explains the larger scattering underestimation with higher BC mass concentration in Fig. 6(a).

To further explore the effects of coating volume changes, Fig. 6(b) shows the size-resolved scattering coefficients before and after composition-averaging for the aerosol populations from scenario 77 at $t = 2$ h, which produced the largest scattering coefficients underestimation ($-32\%$). There is a significant decrease of $\sigma_{\mathrm{scat}}$ in the size range of 400–800 nm in the sensitivity

populations, and the core ratio increment in bin 4 is responsible for this decrease (Fig. S2).

The blue lines in Fig. 7 show the scattering cross sections for two different real refractive indices. For particles with diameters between 800 and 1200 nm, a lower refractive index leads to a larger scattering cross section, although the difference is smaller than the change caused by adding a BC core. Similar to the BC core size change $\Delta D^{\mathrm{core}}$ in Figure 4, we defined a volume-





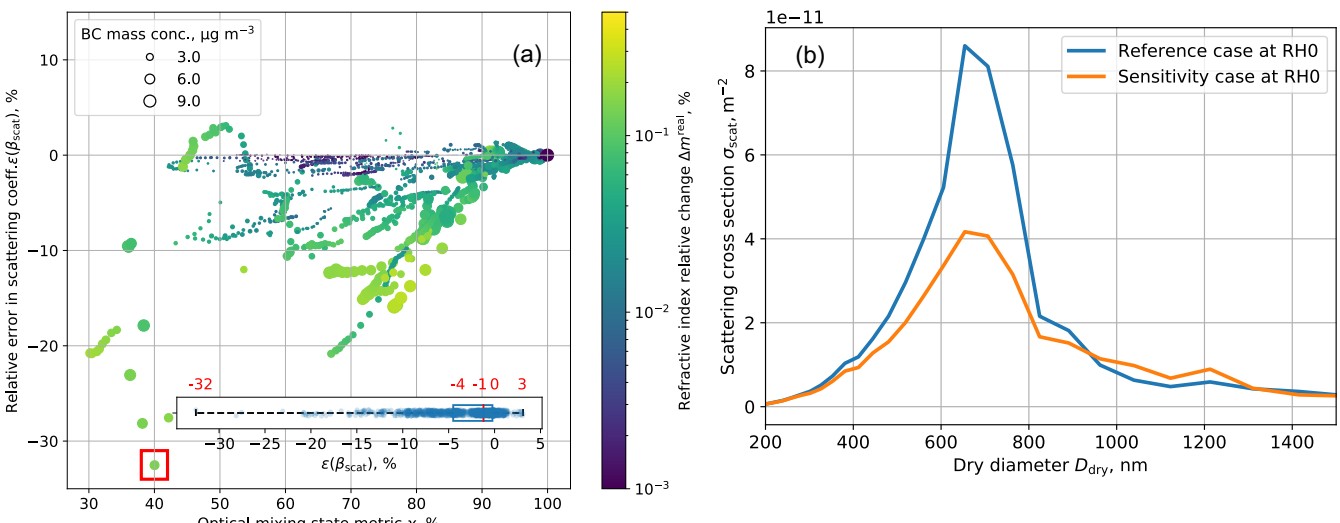

**Figure 6.** (a) Same as Fig. 4, but for $\epsilon(\beta_{\mathrm{scat}})$. The color is for refractive index relative change and the marker size represents BC bulk mass in the population. The red box is the population analyzed in (b). (b) Size-resolved scattering coefficients for the reference and sensitivity (composition-averaged scenario library). The population is from scenario 77 at $t = 2$ h.

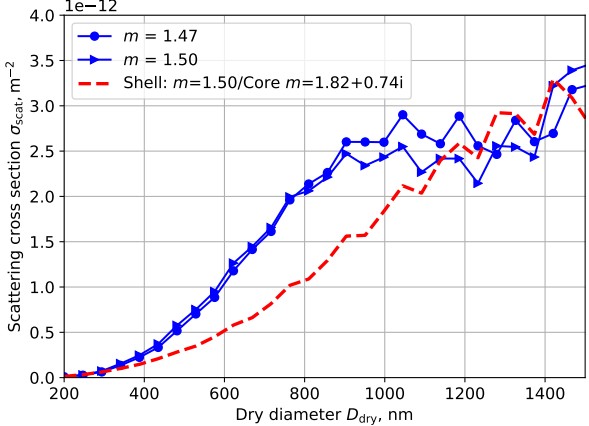

**Figure 7.** Relation between scattering cross section, refractive index and diameter for a wavelength of 550 nm. Blue lines are for non-absorbing particles and symbols indicate refractive index. Red line is for absorbing particles including a BC core of $0.2D$.





weighted refractive index change, $\Delta m^{\mathrm{real}}$, to help understand the changes in scattering. The index change is defined as:

$$\Delta m^{\mathrm{real}} = \frac{\sum_{i=1}^{N} V_i m^{\mathrm{real}\prime} - \sum_{i=1}^{N} V_i m^{\mathrm{real}}}{\sum_{i=1}^{N} V_i m^{\mathrm{real}}},\tag{11}$$

where $i$ is the particle index, $V_i$ is the particle volume, $m^{\mathrm{real}}$ is the real part of the coating refractive index of the particles in the reference library, and $m^{\mathrm{real}\prime}$ is for particles in the sensitivity library. As shown in Fig. 6, aerosol populations with small errors in scattering tend to be associated with small $\Delta m^{\mathrm{real}}$. For more externally-mixed populations (with lower $\chi$), $\Delta m^{\mathrm{real}}$ tended to be larger.

For the effects of composition-averaging for particle scattering, we conclude that at a given value of $\chi$, the magnitude of $\epsilon(\beta_{\mathrm{scat}})$ was determined by the change in core/coating volumes and by changes in the coating refractive index. The increase of BC core sizes after composition-averaging is the major factor for the decrease of the scattering coefficients. Populations with large underestimation are those with higher BC mass concentrations and large refractive index changes. It is worth to emphasize that we did not consider the absorption of organic carbon that might be present in the coating (Esteve et al., 2014).

## 4 The effects of water uptake on aerosol optical properties

The analysis so far was based on dry aerosol populations. In this section we investigate the impact of water uptake on the errors in absorption and scattering by considering RH values of 50% and 90%. As a reminder, we performed composition-averaging on the dry population first, and then calculated water uptake based on the averaged composition for RH=50% and RH=90%, respectively.

Considering all populations, the range of relative errors in $\beta_{\mathrm{scat}}$ decreased with increasing RH, with the median error over all populations decreasing from $-1.2\%$ (RH=0%) to $-1.0\%$ (RH=50%) and $-0.2\%$ (RH=90%) (Fig.8(a)). In contrast, the range of relative errors in $\beta_{\mathrm{abs}}$ remained approximately the same (Fig.8(b)), with a median of approximately 13%.

The different response of $\epsilon(\beta_{\mathrm{scat}})$ and $\epsilon(\beta_{\mathrm{abs}})$ after the populations became humidified was due to the scattering coefficients increasing strongly at higher relative humidities (Fig. S3(a)). The enhancement ratio, defined by the $\beta_{\mathrm{scat}}$ values for the higher RH cases and the dry case, had a median of 1.33 at RH= 50% and 3.35 at RH= 90% in our scenario populations. These values are in accordance with previous studies (Titos et al., 2016; Burgos et al., 2020). As for the absorption coefficients in the humidified environments, the differences between reference and sensitivity cases remained the same (Fig. S3(b)), indicating that the errors in absorptivity introduced by composition-averaging were not sensitive to RH.

## 5 Errors in single scattering albedo and implications for directive radiative forcing

The changes of scattering and absorption coefficients lead to changes in SSA, which is an important quantity that determines radiative forcing. With the definition of SSA, we can calculate the absolute error $\Delta\mathrm{SSA}$ as:

$$\Delta\mathrm{SSA} = \frac{\beta_{\mathrm{scat}}'}{\beta_{\mathrm{scat}}' + \beta_{\mathrm{abs}}'} - \frac{\beta_{\mathrm{scat}}}{\beta_{\mathrm{scat}} + \beta_{\mathrm{abs}}} = \frac{\beta_{\mathrm{scat}}'\beta_{\mathrm{abs}} - \beta_{\mathrm{scat}}\beta_{\mathrm{abs}}'}{(\beta_{\mathrm{scat}}' + \beta_{\mathrm{abs}}')(\beta_{\mathrm{scat}} + \beta_{\mathrm{abs}})},\tag{12}$$



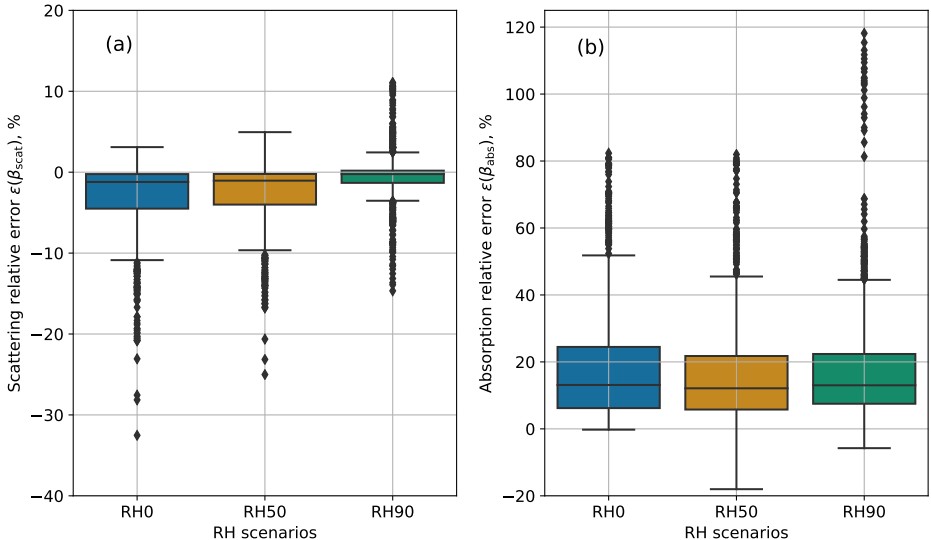

**Figure 8.** Box plot of (a) scattering relative error $\epsilon(\beta_{\mathrm{scat}})$ and (b) absorption relative error $\epsilon(\beta_{\mathrm{abs}})$ at three RH levels (0%, 50% and 90%). Dots are the populations with values outside Q3 + 1.5IQR.

where $\beta'_{\mathrm{scat}}$, $\beta'_{\mathrm{abs}}$ refer to the scattering and absorption coefficients after composition-averaging. Based on the previous analysis, we know that $\beta'_{\mathrm{scat}}$ tends to be lower than $\beta_{\mathrm{scat}}$ and $\beta'_{\mathrm{abs}}$ greater than $\beta_{\mathrm{abs}}$. Combining these changes with equation 12, these variations will result in negative values for $\Delta$SSA and the relative error $\epsilon(\mathrm{SSA})$, which is confirmed by Fig. 9.

Figure 9 shows that $\epsilon(\mathrm{SSA})$ was negative for all the dry aerosol populations, with a median value of $-0.9\%$ and a largest value of $-22.3\%$. The dependence of $\epsilon(\mathrm{SSA})$ on the mixing state metric $\chi$ shows a similar pattern as for the volume scattering coefficient $\epsilon(\beta_{\mathrm{scat}})$. The errors decreased with increasing $\chi$, indicating the SSA underestimation was smaller for more internally mixed populations. For the populations with the same mixing state metric $\chi$, errors were higher for the populations with more BC mass concentrations. Aerosol populations with higher SSA errors were also associated with higher refratcive changes.

In order to further connect $\epsilon(\mathrm{SSA})$ with $\epsilon(\beta_{\mathrm{scat}})$ and $\epsilon(\beta_{\mathrm{scat}})$, and investigate the effects of RH, we sorted the populations by $\epsilon(\beta_{\mathrm{scat})}$ and $\epsilon(\beta_{\mathrm{abs}})$ ranges and calculated the averaged $\epsilon(\mathrm{SSA})$ for each $\epsilon(\beta_{\mathrm{scat}})$-$\epsilon(\beta_{\mathrm{abs}})$ bin for the three RH levels, as shown in Fig. 10. For all three RH levels, $\epsilon(\mathrm{SSA})$ was negative, meaning that composition-averaging causes an underestimation of SSA. The largest $\epsilon(\mathrm{SSA})$ $(-22.3\%)$ occurred for the largest underestimation in $\epsilon(\beta_{\mathrm{scat}})$ in the RH $=0\%$ environment. Populations with $\epsilon(\mathrm{SSA})$ lower than $-10\%$ were related to populations with large negative magnitudes of $\epsilon(\beta_{\mathrm{scat}})$. Relative errors in SSA decreased in a more humidified environment, accompanied by decreasing errors in scattering coefficients. The median underestimation of SSA decreased from $0.9\%$ (RH$= 0\%$) to $0.7\%$ (RH$= 50\%$) and $0.4\%$ (RH$= 90\%$).

The underestimation of SSA can have significant impacts in calculating direct radiative forcing. McComiskey et al. (2008) evaluated the response of directive radiative forcing to changes of several quantities, including aerosol optical depth and single scattering albedo. They found that the total uncertainties in directive radiative forcing ranged from 0.2 to 3.1 $\mathrm{W\,m^{-2}}$, and SSA





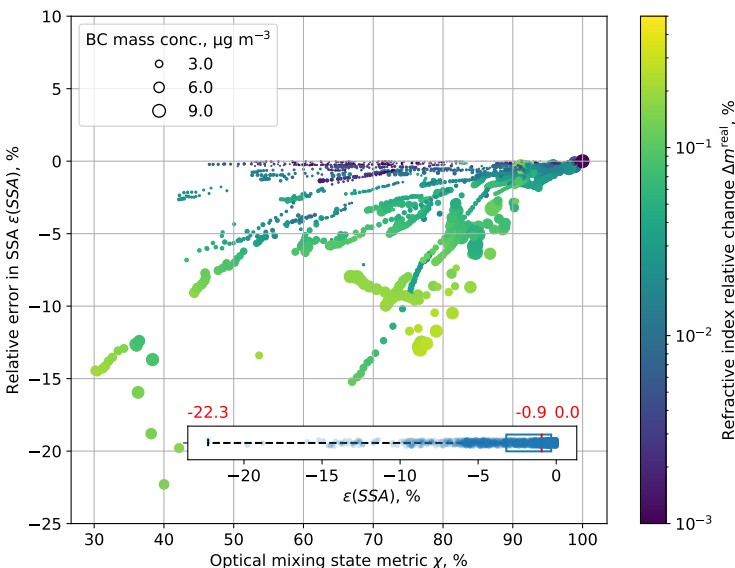

**Figure 9.** Same as Fig. 6(a), but for $\epsilon(\mathrm{SSA})$. Red numbers in the inset box plot are for the minimum, median and maximum values.

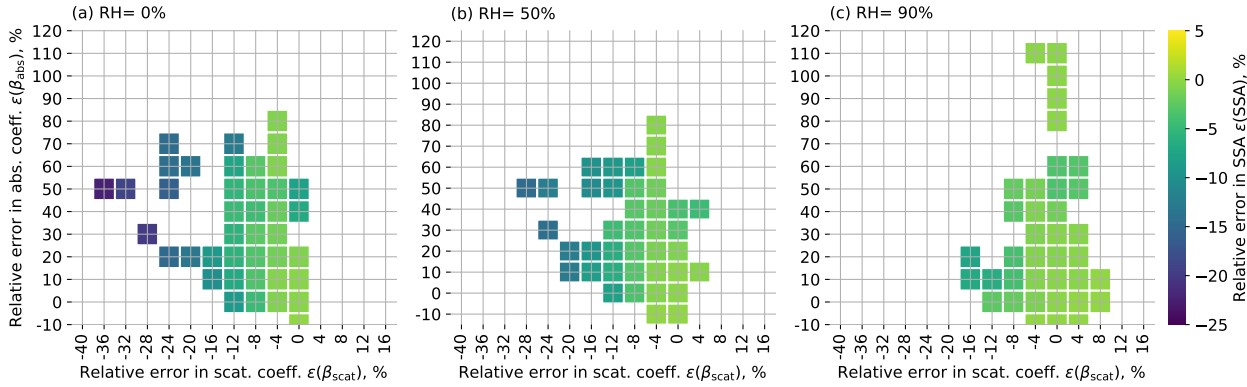

**Figure 10.** Relation between errors in SSA, scattering and absorption coefficients. Color represents the averaged $\epsilon(\mathrm{SSA})$ in the corresponding $\epsilon(\beta_{\mathrm{scat}})$ and $\epsilon(\beta_{\mathrm{abs}})$ histograms.





introduced the largest uncertainties. Through perturbation analysis, Loeb and Su (2010) also found the SSA to be the dominant factor for direct radiative forcing uncertainties. They perturbed SSA by $\pm$ 3% over land, which resulted in uncertainties in direct aerosol radiative forcing between $-0.59$ and $+0.69\ \mathrm{W\,m^{-2}}$. The median SSA errors for our simulations were on the order of the pertubations imposed in the study by Loeb and Su (2010), and we therefore conclude that mixing state effects can

have impacts on radiative forcing similar to the ones reported in Loeb and Su (2010). Furthermore, the spatial and temporal variations of relative humidity imply that the errors in optical properties for a population with a given mixing state may vary depending on location, season and time of day. Radiative transfer calculations would be required for a more in-depth analysis of radiative forcing impacts.

## 6 Conclusion and discussion

Simplified representation of aerosol mixing state used in current regional or global models may introduce errors in simulating aerosol optical properties, thus leading to uncertainties in calculating directive radiative forcing. In this study, the errors introduced by internal mixture assumptions used in sectional aerosol models were systematically quantified. We created a reference scenario library with 1800 aerosol populations by performing particle-resolved aerosol model simulations with PartMC-MOSAIC. We constructed a sensitivity library where particles were internally mixed in a prescribed set of size bins

by applying composition-averaging. This operation has the properties of conserving number concentration and particle sizes, and hence differences in any quantity can be solely attributed to mixing state impacts. Aerosol populations from the reference and sensitivity library were then exposed to three different RH levels to understand the relative role of chemical species and water redistribution introduced by the internal mixture assumption.

The internal mixture assumption generally led to an overestimation of the volume absorption coefficients and an under-

estimation of the volume scattering coefficients. The relative errors for $\epsilon(\beta_{\mathrm{abs}})$ and $\epsilon(\beta_{\mathrm{scat}})$ reached up to 70% and $-32\%$, respectively. The relative errors generally increased for more externally-mixed populations, although at a given value for $\chi$ a range of errors could be found, especially for the error in the scattering coefficient. For the error in the absorption coefficient, this range can be explained by the magnitude of BC core size changes that are induced by composition-averaging. For the error in the scattering coefficient, it can be explained by the magnitude of the changes in the refractive index of the coating that are

induced by the composition-averaging.

For the cases with RH of 50% and 90%, the bulk aerosol water content was almost identical for the aerosol populations in reference and sensitivity libraries. The relative error in the volume absorption coefficient $\epsilon(\beta_{\mathrm{abs}})$ displayed a similar pattern for RH of 50% and 90% compared to the dry environment. The relative error in the volume scattering coefficient $\epsilon(\beta_{\mathrm{scat}})$ decreased for higher relative humidities because of the enhanced scattering cross section through hygroscopic growth.

The absorption overestimation and scattering underestimation resulted in an consistent underestimation of SSA, with median errors of $-0.9\%$ (RH0), $-0.7\%$ (RH50) and $-0.4\%$ (RH90). Populations with the largest underestimation of SSA ($-22.3\%$) were associated with populations with the largest underestimation in scattering.



It is worth emphasizing that we used Mie theory with a core-shell configuration to calculate optical properties assuming spherical particle shapes. Our results are therefore most representative of BC-containing populations where the BC core is collapsed rather than a fractal aggregate (China et al., 2013, 2015). More accurate methods, such as discrete dipole approximation (DDA) should be used to represent these more irregular particle shapes (Scarnato et al., 2013; Curtis et al., 2008; Luo et al., 2019; Wu et al., 2020).

*Code and data availability.* The simulation data and codes availability are accessible at the following: https://doi.org/10.13012/B2IDB-8157303_V1.

*Author contributions.* Y. Yao, Z. Zheng and N. Riemer designed the particle-resolved scenario libraries. J.Curtis developed codes for calculating per-particle optical properties and J. Ching contributed to interpret results. Y. Yao and N. Riemer performed the analysis and prepared the manuscript, with edits from co-authors.

*Competing interests.* The authors declare that there are no competing interests.

*Acknowledgements.* Yu Yao acknowledges funding from the National Science Foundation, Atmospheric and Geospace Sciences NSF AGS grant 1254428. Jeffrey Curtis and Nicole Riemer acknowledge funding from DOE ASR grant DESC0022130. Joseph Ching is an International Research Fellow of Japan Society for the Promotion of Science (JSPS) and acknowledges the financial support from the JSPS Postdoctoral Fellowships for Research in Japan (Standard). Joseph Ching thanks the financial support by Research Institute for Humanity and Nature (RIHN: a constituent member of NIHU, Japan) Project No. 14200133 (Aakash); Arctic Challenge for Sustainability II (ArCS II), grant no. JPMXD1420318865 from the Ministry of Education, Culture, Sports, Science, and Technology (MEXT), Japan; and Fundamental Technology Research (M5 and P5) of Meteorological Research Institute (MRI), Japan. Zhonghua Zheng is funded by the NCAR Advanced Study Program Postdoctoral Fellowship. This material is based upon work supported by the National Center for Atmospheric Research, which is a major facility sponsored by the National Science Foundation under Cooperative Agreement No. 1852977.



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
