# Peer review of "Quantifying the effects of mixing state on aerosol optical properties"

_Atmospheric Chemistry and Physics, 2022_

## Author Comment (AC1)

**Response to Reviewer #1 comments**

Yao et al presents a particle-resolved model study to characterise the relationship between aerosol mixing state and optical properties. A useful mixing state algorithm (mixing state index) is applied to quantify the complicated role of aerosol mixing state in the calculations of aerosol optical properties. Overall, this study is well written, and the results of this study are important for the estimation of atmospheric aerosol climate effects. I have two major comments and several minor comments before the manuscript can be accepted for publication.

We thank the Reviewer for their valuable comments. Changes in response to Reviewers' comments are marked in blue in the revised manuscript.

**Major comments**

(1.1) The aerosol optical properties and mixing state simulated by the particle-resolved model has been discussed well. However, it would be better to present the mass absorption coefficient (MAC) results as well for a broader interest. Given PartMC-MOSAIC can also present mass-resolved results and following the methods described in Fierce et al. (2020), I thzink both the volume-based and mass-based parameters can be derived through the PartMC-MOSAIC simulations.

Yes, the reviewer is right. We can also calculate mass absorption coefficient (MAC) using the model. The following revisions are made to address this:

We included the definition of  $MAC_{BC}$  on p. 7, lines 144–146: "We can also calculated the BC-specific mass absorption coefficients  $MAC_{BC}$  (m2g-1) using

$$MAC_{BC} = \frac{\sum_{i}^{N} \sigma_{abs,i}(\lambda) n_i}{\sum_{i}^{N} m_{BC,i} n_i},$$
(1)

where  $m_{BC,i}$  is the BC mass in particle i."

We also added a description of the MACBC errors on p. 14, line 289: "Since compositionaveraging conserves the bulk species mass concentrations, the denominator in Eq. (5) (total BC mass concentration) remains unchanged, and the errors in MACBC are the same as for  $\beta_{abs}$ ."

Finally, we added a figure of MACBC at different RH levels as Fig. S6(c) and described it on p. 17, line 330–333: "As for the absorption coefficients in the humidified environments, the differences between reference and sensitivity cases remained almost the same for both  $\beta_{abs}$  and MACBC (Fig. S6(b) and (c)), indicating that the errors in absorptivity introduced by composition-averaging were not sensitive to RH."

(1.2) Following the major comment above, it would be helpful to provide the absorption enhancement information (Eabs) as well. I encourage the authors to add the Eabs results as a function of BC mass fraction and include the discussions in the relevant sections.

Thanks for the suggestion. We added the definition of  $E_{abs}$  on p. 7, lines 140–143: "The absorption enhancement of BC-containing particles due to coatings is defined as

$$E_{\rm abs}(\lambda) = \beta_{\rm abs}(\lambda) / \beta_{\rm abs,BC}(\lambda), \tag{2}$$

where  $\beta_{abs,BC}(\lambda)$  is the absorption coefficient when the particle coatings are removed from the BC cores."

We added two new figures, Fig S4 for the correlation between  $E_{abs}$  and BC mass fraction and Fig 4(b) for the correlation between errors in  $E_{abs}$  and mixing state. The related analysis is added on p. 14, lines 285–291: "The coating redistribution after composition-averaging also changes the absorption enhancement. As shown in Fig. S4, the median  $E_{abs}$  is 1.88 for the reference populations with BC mass fraction less than 10%, while it is 1.98 for the corresponding populations of the sensitivity library. The absorption enhancement decreases as the bulk BC mass fraction decreases. These values are within the range of previous studies (Fierce et al., 2020;Cappa et al., 2012). Similar to the error in volume absorption coefficient  $\epsilon(\beta_{abs})$ , the errors are larger for the populations for lower mixing state metric (Fig. 4(b))."

**Minor comments**

(1.3) The authors claimed that the absorption of brown carbon (BrC) is not considered in this study and the maximum diversity value is 2. Therefore, I think the term "chi" mainly works for the BC and non-BC material. I suggest the authors change the term "optical mixing state metrics" to "black carbon mixing state metrics" or just define it as "mixing state metrics".

Thanks for pointing this out. We changed the term "optical mixing state metric" to "black carbon mixing state metric" as suggested on p. 10, line 202 and line 211. We also updated the figure axis labels accordingly.

(1.4) Figure 1: Suggest adding a legend to the figure as colour blue also stands for nitrate in the following graph. The sentence "black stands for black carbon black" also needs rephrasing.

Thanks for pointing this out! Except for black for BC and light blue for water, the other colors in Fig. 1 are just for conceptual illustration and do not represent any particular species (we track 18 aerosol species in addition to BC and water). We revised the wording in the Fig. 1 caption to: "The other colors conceptually represent other chemical species. In total, we track 18 aerosol species in addition to BC and water."

(1.5) Line 196: "in urban environments, BC ages quickly, forming internal mixtures with secondary species". May need a reference for this.

Thanks! We added the reference Riemer et al., 2010 (modeling) and Wang et al., 2010 (observations) on p. 10, line 214–215.

(1.6) Page 13, Line 270: Should be the "core mass ratio" to avoid misleading. As the core volume ratio maintained the same in each bin indicated by the caption of Fig. 7. It might be helpful to add the core mass ratio in each bin to Fig. 7 for better illustration.

Yes, the reviewer is right. We revised the word to "core mass ratio" at p. 16, line 303. As for the core mass ratio in Fig 7, it is the same as the volume mass ratio because we are applying the same density for coating (ammonium nitrate) and core (BC) species, 1800 kg m-3. To make it more clear, we added "constant BC volume fraction across the size range" to the caption.

(1.7) Line 353-357: The authors may benefit from including the results from Hu et al (2021) for the discussions of BC morphology.

Thanks for providing the reference. We added the work of Hu. et al., 2021 as another potential way to improve the shape representation for BC-contained particles at p. 20, line 394.

**Response to Reviewer #2 comments**

This study uses the PartMC-MOSAIC model to evaluate the influence of the treatment of the BC mixing state on aerosol optical properties. The authors show that averaging the mixing state ("composition averaging") overestimates absorption coefficient and underestimates scattering coefficient. In addition, the authors evaluated the dependence of these optical properties on relative humidity.

This study fits well within the scope of the ACP, and their results will be important for more accurate estimation of aerosol optical properties by climate models. The manuscript is generally written well and is suitable for the publication of this journal after considering some minor comments described below.

We thank the Reviewer for their valuable comments. Changes in response to Reviewers' comments are marked in blue in the revised manuscript.

(2.1) Lines 14-19: The authors show some examples of studies estimating direct radiative forcing of BC and aerosols. However, the values in these studies (0.9 W m-2 for BC and -1.9 W m-2 for aerosols) are much larger than the values reported in the IPCC AR6. I suggest the authors revise this part considering the latest findings and assessment reports.

Thanks for pointing this out. We revised the BC and aerosol radiative forcing values to +0.11 and -0.22 (p. 1, lines 16–19), and we added IPCC AR6 Chapter 6 and Chapter 7 as the reference for these values.

(2.2) Lines 52-64: In this paragraph, the authors describe that it is difficult to represent both particle size and mixing state in 3-D models. However, recent studies have developed regional 3-D models and global climate models that explicitly represent both particle size and mixing state (Matsui et al., 2013; Matsui, 2017). They have also evaluated the importance of resolving particle size and mixing state in the estimation of optical properties and radiative forcing (e.g., Matsui and Mahowald, 2017; Matsui et al., 2018). I suggest the authors describe these studies in Introduction or Discussion section.

Thank your for providing these references. We incorporated them into our revised manuscript in the following locations:

The studies of Matsui et al. (2013), Matsui (2017) are now included in the introduction part (p. 3, lines 64–67): "Recently, aerosol modules with more detailed BC mixing state representation were implemented in global climate models (Matsui et al., 2013; Matsui, 2017). These approaches better represent the evolution of BC aging processes in each size bin by adding a second dimension for BC mass fraction. However, this two-dimensional bin approach still does not capture the mixing state information of other, non-BC aerosol species."

The studies of Matsui and Mahowald (2017) and Matsui et al. (2018) are added in the discussion section (p. 19, lines 378–380): "The finding of overestimation of BC absorption due to simplified mixing state representation was consistent with many other studies, including the works by Fierce et al. (2016), Matsui and Mahowald (2017) and Matsui et al. (2018)."

(2.3) Line 107, Table 1: Please show the ranges for model outputs also. It would be good to show how the ranges of mass concentration, number concentration, and mixing state of individual aerosol species in model outputs are consistent with available aerosol observations.

Thanks for the suggestions. Table 1 is only for model inputs, and we documented the model outputs later the paper. Figure 3(a) shows the species concentration ranges and Figure 3(b) shows

the BC mixing state metric  $\chi_{BC}$ . We provide context for our simulated values with observations on p. 10, line 211.

For the revision, we added the distribution of aerosol chemical abundance mixing state metric  $\chi_{\text{chem}}$  to show the mixing state based on individual aerosol species in Figure 3(b) and described it on p. 10, line 211.

We also included Figure 3(d) to show the distribution of total number concentration and added the following information at p. 10, lines 222-226: "The distribution of simulated total number concentration (Fig. 3(d)) are consistent with the observed number concentration of particles in the accumulation mode size range (Asmi et al., 2011). Note that the simulations presented here do not include the process of new particle formation. As a result, the simulated particle populations are more representative of accumulation mode particles in a range of different environments."

(2.4) The caption of Figure 1: black carbon black -> black carbon

Thanks! We corrected it.

(2.5) Lines 160-162: In the composition averaging, the particle sizes of aerosols are also averaged because the resolution is lowered for both mixing state and particle size. How much does the lower resolution of the particle size (particle resolved -> 8 bins) change the results? Can the averaging of the mixing state and the effect of the lower resolution on the particle size be separated?

Thanks for pointing this out. Actually, particle sizes, total number concentration and bulk mass concentration of each species are all preserved within each bin when doing composition-averaging. We only redistribute the species in each particle to make the bin internally mixed. We state this on p. 7, line 154, and cite Ching et al. (2012), where these properties were proven. We modified this statement so that it reads: "The composition-averaging procedure preserves the bulk mass concentration of each species, the total number concentration, and the particle diameters within each bin (Ching et al., 2012, Appendix B1), i.e., after composition-averaging, each bin still contains particles of different sizes."

We added this information in the Fig. 1 legend to remind the reader: "The composition-averaging procedure conserves bulk mass concentration of each species, the total number concentration, and the particle diameters within each bin."

(2.6) Line 180: It is difficult to follow the equations in section 2.5. Can the authors add a figure showing what  $\chi$  means by using the schematic image of particle size and mixing state like Figure 1, for example?

Sorry for the unclear explanation in this section. We added an illustration figure Fig. S1 and related explanations to better explain the variables.

At p. 9 lines 185–187, we added: "As shown in Fig. S1, if a particle only contains one species,  $D_i$  is 1. If the chemical species are present in equal amounts in the particle,  $D_i$  equals the number of species. If the species are unevenly distributed,  $D_i$  is a real number ranging between 1 and the number of species in the particle."

At p. 10 lines 194–199, we added: "Take the three particle populations in Fig. S1 as an example. All three populations have the same bulk species mass concentration. Thus, they have the same bulk effective species diversity  $D_{\gamma}$ . However, the species are distributed differently within the populations. When the particles are externally mixed, each particle only contains one species and  $D_{\alpha}$  is 1, which results in  $\chi = 0\%$ . When all particles have the same species mass fractions,  $D_{\alpha}$  equals to  $D_{\gamma}$ , and we obtain  $\chi$  of 100%. The population is fully internally mixed. For the intermediately mixed population,  $\chi$  ranges between 0% and 100%." (2.7) Line 213, equation 9: Please clarify the difference between v' and v.

We added the description of v' and v on p. 12, lines 238–239: "where v stands for  $\beta_{abs}$ ,  $\beta_{scat}$  or single scattering albedo in the reference library and v' is for same the parameters in the sensitivity library. These parameters are stratified by the mixing state metric  $\chi$ ."

(2.8) Line 222, equation 10: Does  $n_i$  in this equation mean total number concentrations (the sum of particles with and without BC)?

In equation 10,  $n_i$  is the number concentration of particle *i* (i.e., in PartMC each computational particle represents a certain number concentration of simulated particles) and the total number concentration is  $\sum_{i}^{N} n_i$ . If there is no core for particle *i*,  $n_i$  is greater than 0 and  $D_i^{\text{core}}$  is 0. We added the description of  $n_i$  on p. 12, line 249–250: "The number concentration  $n_i$  is always greater than 0, and if there is no core for particle *i*,  $D_i^{\text{core}}$  is 0."

(2.9) Lines 224-228: I think the description that averaging increases BC core particle size is incorrect. As shown on the right side of Figure 5 (at 50%), averaging increases the number of BC containing particles and decreases the BC core diameter of individual BC particles. If I understand correctly,  $\Delta D_{\text{core}}$  in Equation 10 is positive not because BC becomes larger, but because the number of BC containing particles increases (the product of ni and  $Di_{\text{core}}$  is zero for many particles before averaging but is non-zero for all particles after averaging). It would be better to describe that the surface area of particle populations increases, or that the number of BC containing particles increases.

Thanks for the insightful comments. The reviewer is right, after composition averaging, particles with larger diameters experienced a core diameter decrease on a per-particle scale. However, the index  $\Delta D_{\text{core}}$  represents the number weighted average core diameter. For example, for the population at 50% in Fig. 5, the average core diameter of the population is 0.5, and it becomes  $0.793 \ ((0.5)^{\frac{1}{3}})$ . To make it more clear, we added Fig. S2 and refer to this figure on p. 12, line 254: "as shown in Fig. S2."

(2.10) Line 275, equation 11: Do V and m include BC? If so, does this affect the increase in  $\Delta m_{\text{real}}$  because the real part of the refractive index of BC has a larger value than that of other species.

Thanks for pointing this out. In the equation,  $V_i$  is the total volume of particle *i*, including both BC core and coating species. However,  $m^{\text{real}}$  is only for the coating species. Since the total particle volume is preserved while the coating volume changes after composition-averaging, we used the total volume to distinguish the effects of changing coating refractive index. We added the sentences at p. 16 line 310–311, to clarify the motivation of using total volume: "Here we applied total particle volume  $V_i$  in the equation to focus on the relation between the changes in scattering and changes in the refractive index."

(2.11) Line 282, coating refractive index: Related to comment 10, Is  $\Delta m_{\text{real}}$  calculated for coating species only?

The reviewer is right. The  $m^{real}$  is only for coating species.

(2.12) Lines 282-283, The increase of BC core size after composition-averaging: As described in comment 9, this description seems to be incorrect. Please revise.

Please refer to our answers to comment 2.9.

(2.13) Line 310: refractive -> refractive index

Thanks! We revised it (p. 17, line 345).

(2.14) Line 335: As described in comment 5, the averaging does not conserve particle size. Total number and volume (or mass) concentrations are conserved, but surface area and particle size are not necessarily conserved by averaging. I suggest the authors revise this part.

Please refer to our answers to comment 2.5.